# Shock Response Spectrum Analysis of Fatigued Runners

**DOI:** 10.3390/s22062350

**Published:** 2022-03-18

**Authors:** Daniel Benjamin, Serge Odof, Boussad Abbès, François Fourchet, Benoit Christiaen, Redha Taïar

**Affiliations:** 1Podiatry Medicine Department, Centre Luxembourg, 75005 Paris, France; d.benjamin@centre-luxembourg.com (D.B.); b.christiaen@centre-luxembourg.com (B.C.); 2MATériaux et Ingénierie Mécanique (MATIM), Université de Reims Champagne Ardenne, 51100 Reims, France; boussad.abbes@univ-reims.fr; 3École Nationale Supérieure D’ingénieurs de Reims (ESIREIMS), Université de Reims Champagne Ardenne, 51100 Reims, France; serge.odof@univ-reims.fr; 4Physiotherapy Department, Hospital La Tour, 1217 Meyrin, Switzerland; francois.fourchet@gmail.com; 5French Society of Sports Physical Therapist (SFMKS Lab), 93380 Pierrefite sur Seine, France; 6Inter-University Laboratory of Human Movement Biology (LIBM), Savoie Mont-Blanc University, 73000 Chambery, France

**Keywords:** Shock Response Spectrum, fatigue, injuries, gait analysis, Micromachined Microelectromechanical Systems (MEMS) accelerometer

## Abstract

The purpose of this study was to determine the effect of fatigue on impact shock wave attenuation and assess how human biomechanics relate to shock attenuation during running. In this paper, we propose a new methodology for the analysis of shock events occurring during the proposed experimental procedure. Our approach is based on the Shock Response Spectrum (SRS), which is a frequency-based function that is used to indicate the magnitude of vibration due to a shock or a transient event. Five high level CrossFit athletes who ran at least three times per week and who were free from musculoskeletal injury volunteered to take part in this study. Two Micromachined Microelectromechanical Systems (MEMS) accelerometers (RunScribe^®^, San Francisco, CA, USA) were used for this experiment. The two RunScribe pods were mounted on top of the foot in the shoelaces. All five athletes performed three maximum intensity runs: the 1st run was performed after a brief warmup with no prior exercise, then the 2nd and the 3rd run were performed in a fatigued state. Prior to the 2nd and the 3rd run, the athletes were asked to perform at maximum intensity for two minutes on an Assault AirBike to tire them. For all five athletes, there was a direct correlation between fatigue and an increase in the aggressiveness of the SRS. We noticed that for all five athletes for the 3rd run the average SRS peaks were significantly higher than for the 1st run and 2nd run (*p* < 0.01) at the same natural frequency of the athlete. This confirms our hypothesis that fatigue causes a decrease in the shock attenuation capacity of the musculoskeletal system thus potentially involving a higher risk of overuse injury.

## 1. Introduction

Running is the exercise of choice for millions of people all over the world and across the age spectrum. One of the main reasons for its popularity stems from its simplicity. However, running also carries the risk of increased musculoskeletal injuries and there is a need to understand the etiology of injury in order to efficiently prevent it [1]. One of the important functions of the human musculoskeletal system is to attenuate and dissipate shock waves initiated with foot ground contact [2]. Those shock waves are initiated by most types of motion, such as walking and running. The demarcation between walking and running occurs when periods of double support during the stance phase of the gait cycle (both feet are simultaneously in contact with the ground) give way to two periods of double float at the beginning and the end of the swing phase of gait (neither foot is touching the ground) [3]. Generally, as speed increases further, initial contact changes from being on the hindfoot to the forefoot.

Running involves repeated single-leg impacts between the foot and the surface. Such impacts are characterized by a transient peak in the ground reaction force (impact force), rapid deceleration of the lower extremity (impact shock), and the initiation of a wave of acceleration and deceleration (impact shock wave) that is propagated through the body [4].

The impact shock wave experienced by the body due to landings must be attenuated by several structures and mechanisms in the body including bone, synovial fluids, cartilage, soft tissues, joint kinematics and muscular activity. Passively, shock attenuation is achieved by soft tissues and bone. Actively, shock attenuation is achieved through eccentric muscle action [5]. This active mechanism is thought to be far more significant than the passive mechanism in attenuating shock. Since muscles are thought to play a primary role in energy and shock absorption during landing, it has been hypothesized that reduced muscular function, through fatigue, decreases the shock absorbing capacity of the body and subsequently can lead to an increased chance of injury [6]. Fatigue has been defined as any reduction in the force generating capacity of the total neuromuscular system regardless of the force required in any given situation [7].

The loads produced by repeated impacts have been linked to degenerative joint diseases and athletic overuse injuries including, for example, stress fractures, shin splints, osteoarthritis and lower back pain. Although the exact mechanisms of impact related injury are relatively unknown and controversial evidence linking impact, fatigue and injuries are well documented [8,9,10,11].

In this paper, we propose a new methodology for the analysis of shock events occurring during the proposed experimental procedure. Our approach is based on the Shock Response Spectrum (SRS) [12], which is a frequency-based function that is used to indicate the magnitude of vibration due to a shock or a transient event [13]. The main aim is to analyze the ability of the human musculoskeletal system to attenuate the mechanical stresses resulting from the fatigue effect by Shock Responses Spectrum (SRS) of the foot strike–generated shock waves during running. Most previous studies focused on shocks/impacts, ground force reaction, or spectral or vertical impact load rate. Using SRS as a measurement in running gait analysis has never been studied as of today. This innovative approach could pave the way to a whole new way of assessing a runner’s gait pattern using smart connected shoes.

The purpose of this study was to determine the effect of fatigue on impact shock wave attenuation and assess how human biomechanics relate to shock attenuation during running. It was hypothesized that fatigue would cause a decrease in the shock attenuation capacity of the musculoskeletal system, thus potentially involving a higher risk of overuse injury.

## 2. Materials and Methods

### 2.1. Procedures

Five high level CrossFit athletes (four males, one female) who ran at least three times per week and who were free from musculoskeletal injury volunteered to take part in this study. The athletes had a mean age of 26.4 (±3.9) years, stature 182.3 (±5.7) cm, and body mass 81.7 (±8.5) kg, respectively. The athletes usually performed 10 km to 15 km runs twice a week and one sprint interval training of various lengths and intensities. The study was conducted in accordance with the Helsinki Declaration on human experimentation stated in compliance with the 1964 Helsinki Declaration and its later amendments. Every participant provided written consent after information was given on the aim, protocol, and methodology of the study. The original study was approved by the Medical and Ethical Board of the Centre Luxembourg (protocol code LUX_2021_0308_CLAB and date of approval of 3 August 2021). Two Micromachined Microelectromechanical Systems (MEMS) accelerometers (RunScribe^®^) were used for this experiment. The two RunScribe pods were mounted on top of the foot in the shoelaces (Figure 1).

The RunScribe pods encompass 9-Axis Motion Tracking which combines a 3-axis gyroscope, 3-axis accelerometer, and 3-axis compass in the same device together with an onboard Digital Motion Processor. This enables us also to measure at a 500 Hz sampling rate: Efficiency (Stride Rate, Contact Time, Flight Ratio), Motion (Footstrike Type, Pronation, Pronation Velocity), Shock (Impact Gs, Braking Gs), Symmetry and Power.

After a warmup, participants were asked to perform a first 800 m run at maximum intensity. Right after the first run, they jumped on an Assault AirBike (Rogue, Columbus, OH, USA) (Figure 2) where they were asked to perform at maximum intensity for 2 min (the power had to stay above 400 Watts for the 2 min). They dismounted the Assault AirBike and they were then asked again to perform a second 800 m run at maximum intensity. The same protocol was then repeated with another 2 min on the Assault AirBike then a third run at maximum intensity. The RunScribe pods were turned off during the Assault AirBike sessions and were only recording the three 800 m run intervals.

The Assault AirBike, also known as “the Devil’s tricycle” was used to induce fatigue because they procure a unique and extremely challenging effort. It is considered among the crossfit community as the most dreaded but most effective tool for HIIT (High Intensity Interval Training) and metabolic conditioning.

### 2.2. Shock Response Spectrum Calculation

Spectral analysis is commonly used to study the structure of composite waveforms, such as the impact shock waves. The primary tool of spectral analysis is the Fast Fourier Transformation (FFT) that enables us to determine the runner’s natural frequency [14] which corresponds to the peak of the Power Spectral Density (Figure 3):(1)$$PSD=\frac{1}{N}{\left|\int_{-\infty }^{+\infty }a\left(t\right)e^{-j2\pi ft}dt\right|}^{2}$$
where N is the number of points of the recording, a(t) is the acceleration modulus, f is the frequency and t is the time.

Power Spectral Density (PSD) provides a convenient method for separating different frequency components in the impact shock wave, such as acceleration moments due to impact shock [15].

In this paper, we propose a new methodology for the analysis of shock events occurring during the proposed experimental procedure. Our approach is based on the Shock Response Spectrum (SRS), which is a frequency-based function that is used to indicate the magnitude of vibration due to a shock or a transient event. The following procedure, consisting of several steps, is adopted in the present study:Step 1:

The acceleration modulus a(t) is extracted from the recording. Figure 4 illustrates the acceleration modulus results for one CrossFit athlete.

Step 2:

The power spectral density (PSD) given in Equation (1) is then calculated using a Fast Fourier Transform (FFT). This calculation allows us to determine the fundamental frequency of the runner $f_{0}$ corresponding to the position of the largest peak of the PSD. The inverse of this frequency gives the time period of the runner’s step as: $T=1/f_{0}$. The proposed algorithm extracts automatically the “first” step from the entire signal, and thus defines the “pattern” of the runner as shown in Figure 5.

Step 3:

We then carry out the cross-correlation CC(τ) between the runner’s pattern and the recording’s duration a(t):(2)$$CC\left(\tau \right)=\int_{-\infty }^{+\infty }a\left(t\right)pattern\left(t+\tau \right)dt$$

We observe that at each step, the convolution is maximum. For each maximum value of CC(τ) we calculate the SRS of each step and of the entire signal as explained in the next step.

Step 4:

The calculation of the SRS is based on the acceleration time history. It applies an acceleration time history as a common base excitation $\left(\ddot{y}\right)$ to an array of single-degree-of-freedom (SDOF) systems composed of spring $\left(k_{i}\right)$, mass $\left(m_{i}\right)$ and damper $\left(d_{i}\right)$, as depicted in Figure 6.

${\ddot{x}}_{i}$ is the absolute response of each system to the input $\ddot{y}$. This can be determined by applying Newton’s law to a free-body diagram of an individual system, as shown in Figure 7.

The force balance yields the following governing differential equation of motion:(3)$$m\ddot{x}+d\dot{x}+kx=d\dot{y}+ky$$

By defining the relative displacement z=x−y, Equation (3) can be rewritten as:(4)$$\ddot{z}+2\xi \omega \dot{z}+\omega^{2}z=-\ddot{y}$$
where ω0=km is the natural frequency in radians per second and ξ=d(2ω0m) is the damping ratio. Moreover, ξ is usually represented by the amplification factor Q=1(2ξ).

Since the base excitation $\ddot{y}$ is an arbitrary function of time, Equation (4) does not have a closed-form solution. To calculate the SRS of each step and of the entire signal, we have used the algorithm for the calculation of the SRS proposed in [13]. SRS enables us to determine the maximum acceleration a system will undergo when one knows the natural frequency $f_{0}$ and the quality factor Q for each possible natural frequency. In this study, a relative damping of 5% was used, resulting in Q=10. SRS can also be calculated for the entire duration of a recording. We then observed the peaks at the fundamental and harmonic frequencies of the recorded signal [16]. In this context, SRS combines both the notion of transfer function and response to transient regimes.

Intra comparison of the SRS offers a lot more finesse to the analysis since the frequency is also taken into account. The aggressiveness of a running step is not only due to the value of the maximum acceleration but also to the general shape of the movement, only the SRS allows this to be taken into account in the analysis.

Figure 8 gives the general workflow for SRS determination.

## 3. Results

A goal of the present study was to analyze the effect of fatigue through SRS on the ability of the human musculoskeletal system to attenuate foot strike–generated shock waves. The results of this study suggest that, for the analysis of impact shock during running, the different components of the acceleration signal can be distinguished in the frequency domain by means of spectral analysis as shown in Figure 9.

The main advantage of spectral analysis over time-domain analysis of the impact shock wave is the ability to separate spectral peaks from the rest of the data. Since the motion, impact, and resonant components of the acceleration signal have different fundamental frequencies: they produce peaks at different points in the power spectrum [12].

The hypothesis is that fatigue hampers the ability of the human musculoskeletal system to protect itself from overloading due to foot strike–generated shock waves, loss of protection may manifest as an increased shock wave amplitude. For all five athletes, there was a direct correlation between fatigue and an increase in the aggressiveness of the SRS as shown in Figure 10. We noticed that for all five athletes for the 3rd run the average SRS peak was significantly higher than for the 1st run and 2nd run (*p* < 0.01) at the same natural frequency of the athlete. This confirms our hypothesis that fatigue causes a decrease in the shock attenuation capacity of the musculoskeletal system thus potentially involving a higher risk of overuse injury.

When fatigue begins, we could hypothesize that athletes will slow down as a protective means. The result could be moving away from the state of fatigue, in which case the acceleration data could have not increased. It was not the case in our study.

Previous studies have shown that the loading rate of the lower limb is directly and highly correlated with running speed, and the vertical impact force increased with increasing running velocity [17].

Muscle activation lowers the bending stress on bone and attenuates the peak dynamic loads that can damage musculoskeletal tissues. Previous studies have suggested that the fatigued muscles cannot support “optimal” running and they also suggested that fatigue of the runner may lead to modification of landing phase mechanics. It was also found that the transfer of mechanical energy between the eccentric and concentric phases is drastically reduced during muscle fatigue. Such changes may be involved in the development of injuries [18,19,20].

## 4. Discussion

According to the results presented in this study, for the acceleration data to increase, fatigue should be present. We may conclude that the musculoskeletal system becomes less capable of handling foot strike–induced shock waves when the muscles are significantly fatigued. One of the most common running overuse injuries are bone stress fractures (SF) [21,22]. In bones, microcracks are normally present and are thought to be fatigue-related cracks because their numbers increase following repetitive loading. Bone remodeling serves to repair fatigue microcracks. When a bone is loaded repeatedly, resulting in repetitive or cyclic strain, the subsequent accumulation of microdamage is believed to be the threshold of a pathological continuum that is clinically manifested as stress reactions and SF (29). Ultimately, if the activity is not ceased and the bone is not able to self-repair, a complete bone fracture might ensue. Notably, with increasing strain or greater strain rates, the number of loading cycles a bone–29 can withstand before a fatigue failure occurs is reduced [23]. Stress fractures are the clinical manifestation of the accumulation of fatigue damage in bones [24,25,26]. Although the effect of running and its mechanical strain in bone tissues is well documented, the evidence for SF etiology is less conclusive [24,27,28,29,30]. Nevertheless, several researchers reported clear relationships between bone stress related injuries and fatigue. For instance, it is known that the tensile strains on the tensile side of a bending bone are dampened by the contraction of adjacent muscles, aiming at protecting the bone from stress related injury [29,30,31,32,33,34]. It may then be hypothesized that muscles also play the role of shock absorbers and that consequently, muscle fatigue might decrease their absorption properties, resulting in a more aggressive loading rate or loading peak at the bones as fatigue increases [31,32,33].

The obtained results showed that acceleration amplitude steadily increased with the fatigue group and that there was a clear association between fatigue and shock waves (as revealed by the SRS). We may then confirm the conclusions of the aforementioned studies, that the human musculoskeletal system becomes less capable of single leg strike–induced shock waves absorption when the muscles are significantly fatigued. This condition may promote the development of injuries and the present results have a significant implication regarding the etiology of running injuries. Therefore, several recommendations may be effective towards runners’ community or coaches in order to reduce this stress related injury risk, notably as proposed by the multifactorial model of Brukner and Khan [35].

First, it may be advantageous to ensure that the majority of training and exercise is performed to avoid severe fatigue and in line with the load management theory. For instance, external parameters must be considered, such as progressive increment of training loads [36], training surfaces or footwear adaptations [35]. Understanding the influence of SRS on fatigue and on the magnitude of dynamic loading on the human musculoskeletal system will allow the development of proper training procedures and may participate in the reduction of damages to the musculoskeletal tissues.

Secondly and directly in accordance with the present research purpose, lower limb muscles resistance to fatigue is a major component of stress related injury prevention in runners. The present outcomes are in line with former findings reporting that fatigue-related imbalance between the plantar flexors and dorsiflexors may compromise the protective action of these muscles on the lower leg bony structures [29]. Here, it is plausible that deteriorated properties of the calf muscles due to fatigue may affect the role of these soft tissues to protect the bone from stress injury risk.

Finally, injuries in running are also often provoked by fatigue and improper technique, which are both reflected in the runner’s kinematics [18,37,38]. A gait retraining approach has been proposed by several researchers through a modest increase in step rate or a transition from rearfoot to forefoot strike and was considered as effective notably at reducing impact forces and vertical load rate and then at preventing running-related bone stress injuries [37,39].

An individualized approach is nevertheless of high interest and most likely available nowadays. Indeed, state-of-the-art research on kinematics in sports uses optical motion capture systems that are inaccessible to most athletes. With the recent development of Micromachined Microelectromechanical Systems (MEMS), inertial sensors have become widely used in the research of wearable running gait analysis [40] due to several factors, such as being easy-to-use and low-cost. Considering the fact that each individual has a unique way of running, inertial sensors can be applied to the problem of gait recognition where assessed gait can be interpreted as a biometric signature. Thus, inertial sensor-based gait recognition has a great potential to play an important role in many health-related applications. In this work, we demonstrated the potential of wearable technology for the assessment of kinematic parameters using the example of running. We concluded that wearable technology opens possibilities for technique improvement and injury risk reduction to a wide spectrum of athletes. Since inertial sensors are included in smart devices that are nowadays present at every step, inertial sensor-based gait recognition has become a very attractive and emerging field of research that will provide many interesting discoveries.

Although the small sample size is indeed a limitation to applying our findings to the general population, this study is a qualitative and prospective research study exploring a novel and unknown topic. Using SRS as a measurement in running gait analysis has never been studied as of today, leaving us with very little data similar to our study design to be able to calculate a traditional sample size. However, these results still provide valuable information regarding the use of SRS as a biomechanical risk factor in runners. Larger studies on this topic will further advance our understanding of injury risk in runners. It is acknowledged that the relatively low subject numbers used in this study limit the drawing of definitive conclusions, this is particularly true if the study findings conflict with those of previous investigations. The results of our research were in accordance with previous studies and our hypothesis. In the future, a study on a larger group of athletes will be carried out to confirm our previous findings. It is also planned to carry out the same type of studies on high level runners and compare results. Our obvious hypothesis is that elite runners will have a unique ability to dampen the SRS and/or sustain a much higher SRS threshold.

## Figures and Tables

**Figure 1 sensors-22-02350-f001:**
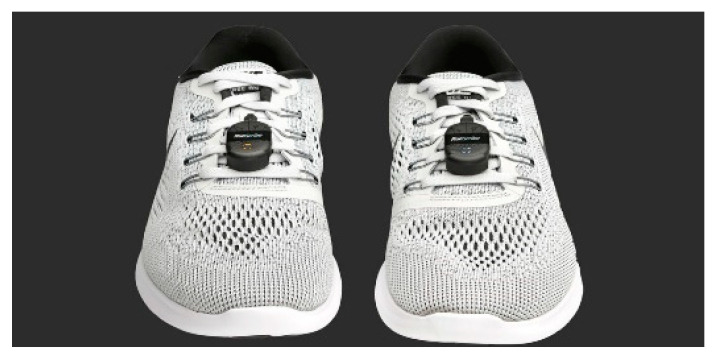
MEMS accelerometers placement.

**Figure 2 sensors-22-02350-f002:**
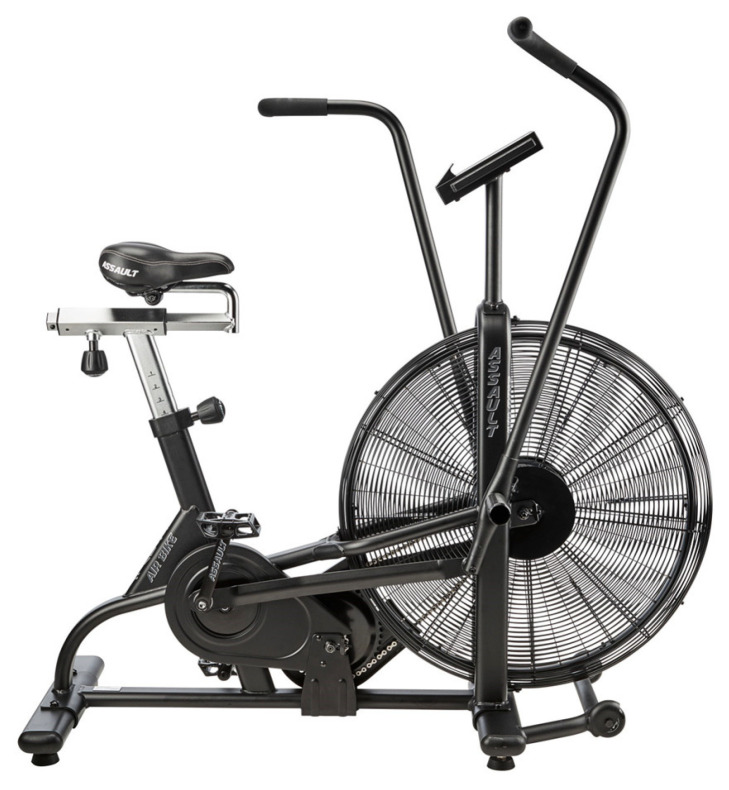
Assault AirBike.

**Figure 3 sensors-22-02350-f003:**
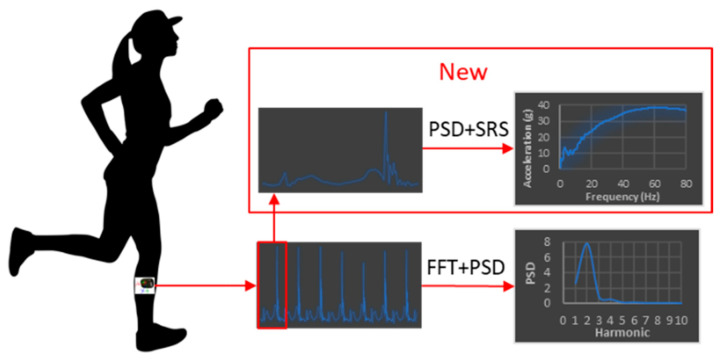
SRS vs. PSD running analysis.

**Figure 4 sensors-22-02350-f004:**
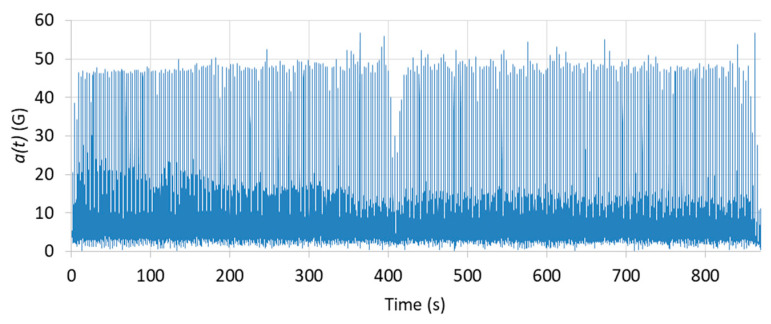
Acceleration modulus a(t) for one CrossFit athlete.

**Figure 5 sensors-22-02350-f005:**
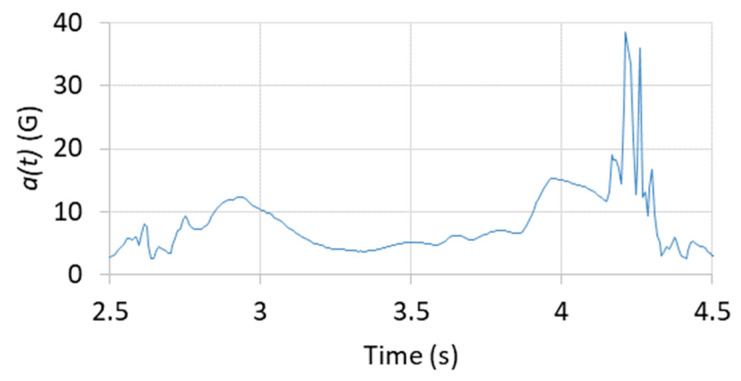
Pattern of one CrossFit athlete.

**Figure 6 sensors-22-02350-f006:**
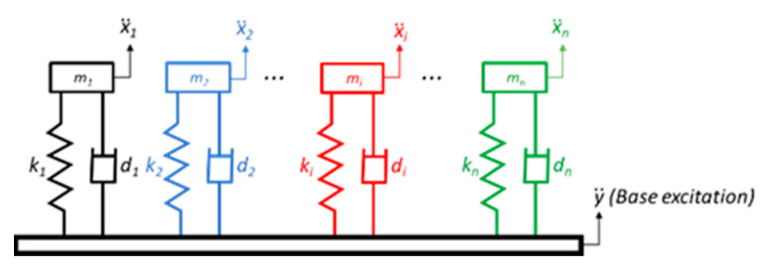
SRS model.

**Figure 7 sensors-22-02350-f007:**
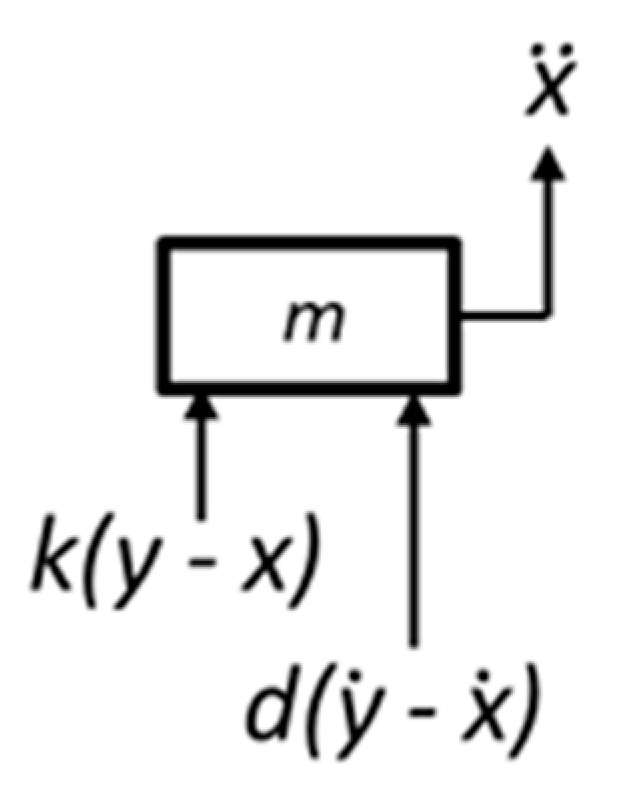
Free-body diagram of an individual system.

**Figure 8 sensors-22-02350-f008:**
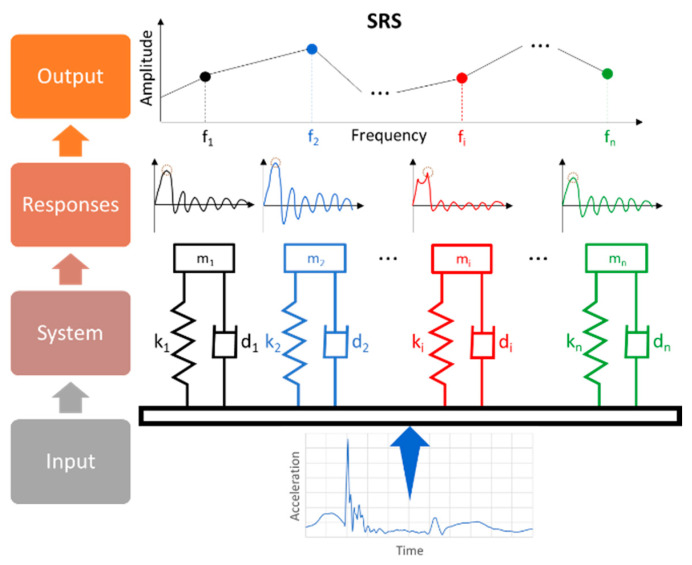
Workflow for SRS determination.

**Figure 9 sensors-22-02350-f009:**
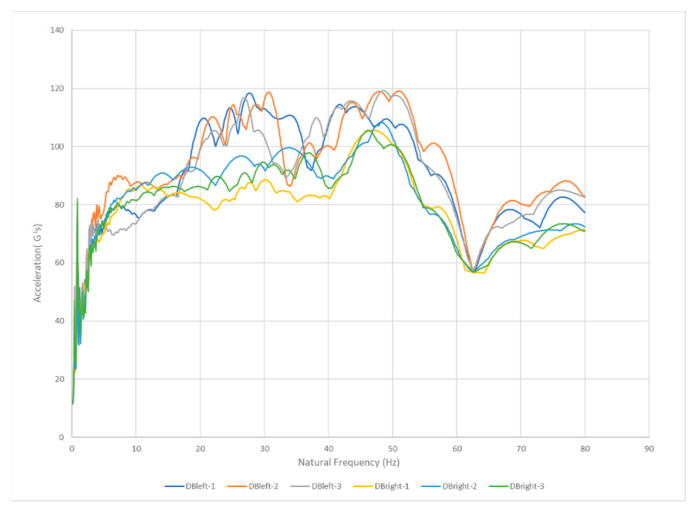
Example of SRS results for one CrossFit athlete extracted for three runs on both feet.

**Figure 10 sensors-22-02350-f010:**
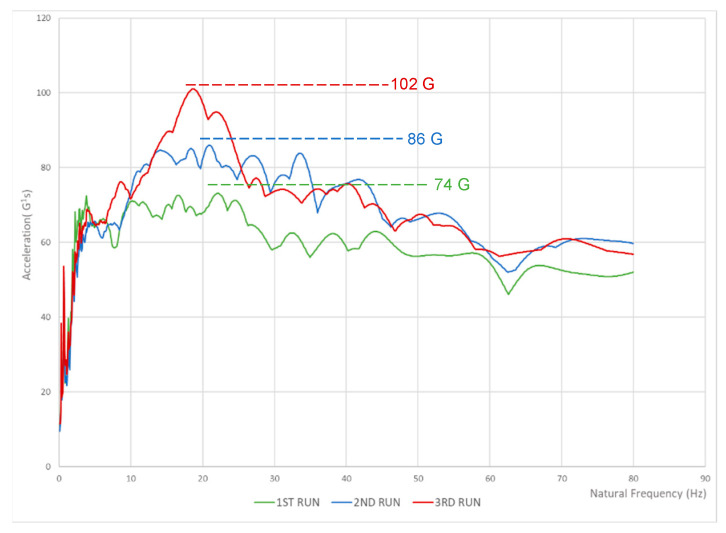
Average SRS peaks for every athlete.

## Data Availability

The data presented in this study are available on request from the corresponding author. The data are not publicly available due to ethical reasons.

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
