# Peer review of "Shock Response Spectrum Analysis of Fatigued Runners"

_sensors, 2022, doi:10.3390/s22062350_

Round 1

Reviewer 1 Report

An interesting article on a so far unknown topic.

I have a few comments:

Materials and methods:

Why research only on 5 people? A larger group is recommended for evaluation.

Discussion:

You have to add weak points of work. It is certainly necessary to write that the study was carried out on a small group of people - only 5. In the future, a study on a larger group of patients is planned.

Please add a few clear conclusions

Author Response

Dear Reviewer,

We appreciate the kind consideration of this paper by the reviewer and the opportunity to highlight the value of our research. The paper has been improved as detailed below, based on reviewer’s thoughtful comments. We have noted our changes in “Bleu” so they are easy to locate. We thank you sincerely for your time and consideration.

Reviewer 2 Report

REFEREE’S REPORT FORM

 Title: Shock Response Spectrum Analysis of Fatigued Runners

Article Type: Research Paper

Comments for transmission to authors

General comments:

This paper has a potential to be accepted, but some important points have to be clarified or fixed before we can proceed and a positive action can be taken.

The points are summarized below:

  1. Why the authors chose as subjects high level CrossFit athletes to study running fatigue and not high level runners? The research literature indicates that certain physiological differences and adaptations exist between these two athlete populations.
  2. The sex of the subjects is missing.
  3. How intensity runs were defined and assessed?
  4. How fatigued state was assessed?
  5. The assessed muscle fatigue was central, peripheral or a combination of central and peripheral fatigue?
  6. Subjects are referred sometimes as high level CrossFit athletes and other times as runners,
  7. How maximum intensity was defined and assessed?
  8. Watts, speed and RPM at maximum intensity on the Assault AirBike console should be given in method section.
  9. The authors state that “... Five high level CrossFit athletes who ran at least 3 times per week”. They do not, however, provide enough information on how long they have been running, as well as on the intensity, distance and time of their runs.
  10. The word “Power” in line 107 needs to be specified.
  11. Sentences/Statements in lines 221-227 need references.

Author Response

(The authors gave the same response as above.)

Reviewer 3 Report

This is a well written article. However prior publication adding some additional comments for clarity and to improve the generalization of results. 

Introduction

Can you add information that links the SRS and injury predition? Providing the threshold of injury or fatigue onset would be very interesting.

Methods

Why was only 5 participants used? 

What was the reasoning for your fatuque protocol? I am sure there is other research where fatigue protocols are provided.

Why not use a longitutdinal design i.e. continuous monitoring with testing the SRS every 1 minute, during the running?

Did you measure the maximum intensity to ensure the participants did it properly?

Results

There are many ways to compare the simularity of time-series data such as shown in these studies

https://dl.acm.org/doi/10.1145/2207676.2208556

https://www.sciencedirect.com/science/article/abs/pii/S0957417421006503 

By using these methods a statistical method of analysis could then be applied to strengthen the conclusions of the article. 

Discussion

With only 5 participants it is hard to provide your conclusions without a section highlighting the limits of this study. 

Author Response

(The authors gave the same response as above.)
